PipeCoV: a pipeline for SARS-CoV-2 genome assembly, annotation and variant identification

http://orcid.org/0000-0002-0111-8209 Oliveira Renato R. M. 1 2 renato.renison@gmail.com
Costa Negri Tatianne 1 tatianne.negri@pq.itv.org
Nunes Gisele 1
Medeiros Inácio 3
Araújo Guilherme 3 4
de Oliveira Silva Fabricio 1
Estefano Santana de Souza Jorge 3 4
Alves Ronnie 1 5
http://orcid.org/0000-0003-0054-3438 Oliveira Guilherme 1
1 Environmental Genomics, Instituto Tecnológico Vale , Belém, Pará , Brazil
2 Programa de Pós-Graduação em Bioinformática, Universidade Federal de Minas Gerais , Belo Horizonte, Minas Gerais , Brazil
3 Programa de Pós-Graduação em Bioinformática, Universidade Federal do Rio Grande do Norte , Natal, Rio Grande do Norte , Brazil
4 Bioinformatics Multidisciplinary Environment, Universidade Federal do Rio Grande do Norte , Natal, Rio Grande do Norte , Brazil
5 Programa de Pós-Graduação em Ciência da Computação, Universidade Federal do Pará , Belém, Pará , Brazil
NICOLÁS MARISA
Electronic publication date: 2022 Apr 13
Publication date: 2022
Volume: 10
Electronic Location ID: e13300
Received 2021 Nov 22; Accepted 2022 Mar 29
Copyright: © 2022 Oliveira et al.
Copyright year: 2022
Copyright holder: Oliveira et al.
License: This is an open access article distributed under the terms of the Creative Commons Attribution License, which permits unrestricted use, distribution, reproduction and adaptation in any medium and for any purpose provided that it is properly attributed. For attribution, the original author(s), title, publication source (PeerJ) and either DOI or URL of the article must be cited.
License URL: https://creativecommons.org/licenses/by/4.0/

Keywords: Sarscov2, Genomics, Variant identification, Covid19, Pipeline, Annotation, Virus

Funding: Vale (Projeto Covid) RBRS000603.12 CABANA RCUK/BB/P027849/1 CNPq (Conselho Nacional de Desenvolvimento Científico) 307479/2016-1 Tatianne Costa Negri is a Fiocruz VPPCB−007−FEX−20 This work received financial support from Vale (Projeto Covid, RBRS000603.12) and the CABANA project (RCUK/BB/P027849/1) to Guilherme Oliveira. Guilherme Oliveira is a CNPq (Conselho Nacional de Desenvolvimento Científico) fellow (307479/2016-1), Tatianne Costa Negri is a Fiocruz fellow (VPPCB-007-FEX-20). The funders had no role in study design, data collection and analysis, decision to publish, or preparation of the manuscript.

==============================
Motivation

Since the identification of the novel coronavirus (SARS-CoV-2), the scientific community has made a huge effort to understand the virus biology and to develop vaccines. Next-generation sequencing strategies have been successful in understanding the evolution of infectious diseases as well as facilitating the development of molecular diagnostics and treatments. Thousands of genomes are being generated weekly to understand the genetic characteristics of this virus. Efficient pipelines are needed to analyze the vast amount of data generated. Here we present a new pipeline designed for genomic analysis and variant identification of the SARS-CoV-2 virus.

Results

PipeCoV shows better performance when compared to well-established SARS-CoV-2 pipelines, with a lower content of Ns and higher genome coverage when compared to the Wuhan reference. It also provides a variant report not offered by other tested pipelines.

Availability

https://github.com/alvesrco/pipecov.

Introduction

Many scientific challenges have arisen in the most different areas of science since the first occurrence of SARS-CoV-2 in December 2019. In bioinformatics, it was no different since thousands of genomes of the virus have been generated globally since the World Health Organization (WHO) declared the coronavirus disease (COVID-19) as a global pandemic. Information contained in the virus genome supported the development of diagnostics based on the polymerase chain reaction (PCR), identification of circulating strains, detection of mutations that can affect the rates of transmissibility, pathogenicity, the study of drugs, culminating with the delivery of vaccines (Mercer & Salit, 2021). Until January 2022, a total of 7,706,888 Sars-Cov-2 genomes have been deposited in EpiCov (Shu & McCauley, 2017), a reference repository for COVID-19 genomic data. Several pipelines for the analysis of the SARS-CoV-2 virus, such as genome assembly (Bedford et al., 2020; Schubert, Lindgreen & Orlando, 2016), variant analysis (O’Toole et al., 2020), the identification of mutations and deletions (Rahman et al., 2020; Desai et al., 2021), and phylogenomics (Forster et al., 2020), among others, were developed.

Two main strategies are used for assembling the genome: reference-based or de novo assembly. Reference-based assembly is performed if the genome of the target organism is available, and the reads are aligned to the reference. This approach focuses more on the structure and the genes found, accepting the loss of indels or multiple nucleotides variant information (Chen et al., 2017). The de novo assembly is performed without the need for a reference genome, by using heuristics to generate consensus sequences and maintaining the single/multiple nucleotide variants and indels (Li, 2012).

Most of the pipelines developed for the genome assembly of SARS-CoV-2 follow a reference-based strategy. Viralrecon (Patel et al., 2020) was built to perform assembly and intra-host/low-frequency variant calling for viral samples. This pipeline supports both Illumina and Oxford Nanopore sequence data and uses Docker/Singularity containers, making installation easy and results highly reproducible. V-pipe (Posada-Céspedes et al., 2021) uses the workflow management system Snakemake to determine the order in which the steps of the specified pipeline are executed with further verification output files. To simplify the installation of all components, conda environments are provided. SIGNAL (Nasir et al., 2020) is a standardized workflow that processes the assembly of short-read viral sequences, compatible with the Illumina ARTIC pipeline and producing the consensus and variants using Ivar (Castellano et al., 2021). SIGNAL contains additional quality control steps and visualization tools, including an interactive HTML summary of the run results. Finally, QIAGEN has released example workflows and tutorials for analyzing Illumina and Oxford Nanopore SARS-CoV-2 sequence data using “CLC Genomics Workbench v20.0.3” (Insights, 2020). These workflows are “Research Use Only” (RUO), and may need to be modified to fit upstream protocols. Free temporary licenses for CLC (Insights, 2020), GWB and IPA are available for testing. Viralrecon, V-pipe, SIGNAL, and CLC pipelines use only reference-based assembly and may be missing critical information regarding multiple nucleotide variants or indels by forcing the resulting consensus to have the same structure present in the reference.

On the other hand, de novo assemblies aim to capture the natural structure of the genomes by maintaining any indel or multiple nucleotide variants found in the sequences. Until now, there was no published pipeline capable of combining reference-based and de novo assembly strategies for the SARS-CoV-2 genome. Such a combination of strategies could aggregate valuable and reliable information from an analyzed virus sample since it would recover natural genome information such as indels and multiple nucleotide variants and also use the reference genome to first guide and organize the de novo assembled sequences.

To combine both advantages of using de novo and reference-based assembly strategies, we present PipeCoV, an open-source and efficient pipeline that provides assembly to variant identification of SARS-CoV-2 viruses. PipeCoV consists of several steps: (i) read quality control (Fig. 1A), (ii) reference mapping (Fig. 1B), (iii) de novo genome assembly (Fig. 1C), (iv) reference-based assembly (Fig. 1D), (v) gap closing (Fig. 1E), (vi) annotation (Fig. 1F), and (vii) variant identification (Fig. 1G). All the steps were consolidated into two command lines offering efficient and fast results. To facilitate usage, the pipeline was compiled into docker containers (Merkel, 2014) to avoid problems with incompatibility and installation of the tools. PipeCoV was validated using 120 amplicon datasets extracted from the covid data portal (https://www.covid19dataportal.org/) under Bioproject PRJNA691556 (Gupta et al., 2021) and compared to Viralrecon, V-pipe, Signal, and CLC.

Figure 1 The PipeCoV workflow.

The figure shows all steps and software used by PipeCoV. (A) Preprocessing, trimming, and removal of low-quality reads. (B) Mapping reads using a reference genome. (C) D. novo genome assembly. (D) Reference-based assembly. (E) Gap closing. (F) Genome annotation using PROKKA Database. (G) Variant identification by Pangolin software.

Methods

We adapted the pipeline developed by Bedford et al. (2020), allowing the analyses of both shotgun and amplicon datasets from paired-end sequencing. After sequencing, AdapterRemoval v2.2.3 (Schubert, Lindgreen & Orlando, 2016) removes adapters from sequences (−a), using a text file containing the adapters sequences, trims low-quality bases, and discards reads with score quality less than a given Phred score (−q 20) and smaller than a given length (−l 100). PipeCoV generates a quality report before and after AdapterRemoval (Schubert, Lindgreen & Orlando, 2016), using FastQC v0.11.9 (Andrews, 2010). All these steps (Fig. 1A) are included in the qc_docker.sh script and can be run with the following command line (The default parameter values are described in the example below):

qc_docker.sh -i illumina -1 SAMPLE_R1.fastq -2 SAMPLE_R2.fastq

-a adapters.txt -q 20 -l 100 -o output_qc -t 24

Help commands:

-i is the sequencing instrument;

-1 is the forward reads;

-2 is the reverse reads;

-a is the text file containing the adapters sequences;

-q trim low-quality bases and discard reads with quality scores less than a given Phred score;

-l shorter than the given length;

-o is the output folder;

-t is the number of threads to be used;

The following steps are included in the assembly_docker.sh script. For decontamination, BBDuk v38.76 (Bushnell, Rood & Singer, 2017) maps high-quality reads to a virus reference genome (−r) with a kmer size (−k) and maximum mismatch (−m) chosen by the user. The high-quality reads are also mapped to the reference genome by Bowtie2 v2.3.5.1 (Langmead & Salzberg, 2012) and Samtools (Li et al., 2009), with default settings, to generate the first ordered BAM file (Fig. 1B).

All reads mapped to the virus reference genome will are de novo assembled using SPAdes v3.15.0 (Bankevich et al., 2012) on metagenomic mode (–meta), with a maximum memory defined by the user (−g) and the remaining default settings (Fig. 1C). Then, the hcov_make_seq.R script (Bedford et al., 2020) generates the first consensus sequence, using BWA v0.7.17 (Li & Durbin, 2009) and Samtools (Li et al., 2009) to align the resulting contigs to the reference genome, selecting contigs bigger than a minimum length (−l) and with higher coverage (−c). The mapped reads will also be aligned to the first consensus sequence by Bowtie2 (Langmead & Salzberg, 2012) and Samtools (Li et al., 2009), generating the second ordered BAM file. Both first and second ordered BAM files will be used by hcov_generate_consensus.R script (Bedford et al., 2020) to generate an intermediate consensus sequence using the virus reference genome (Fig. 1D). In this previous step, the de novo assembly meets the reference assembly, using the contigs assembled by SPAdes (Bankevich et al., 2012). The SPAdes generated contigs are aligned to the reference and used by the hcov_make_seq.R script to generate the first consensus that is polished and corrected by the two ordered BAM files. With this combination of strategies, any information obtained in a de novo assembly using SPAdes (Bankevich et al., 2012) is maintained along with the reference assembly, generating a robust intermediate consensus. The gaps in this intermediate sequence are filled by GapCloser v1.12-r6 (Luo et al., 2012) using the default parameters, generating a final consensus sequence (Fig. 1E), which is then annotated by Prokka v1.14.5 (Merkel, 2014) (–kingdom Viruses –genus Betacoronavirus) (Fig. 1F). Lineage identification is performed by Pangolin v2.3.8 (O’Toole et al., 2020) with default parameters (Fig. 1G). Each tool used on PipeCoV is encapsulated into docker containers, and at each run, the used containers are removed from the environment. This permits configuring PipeCoV to always use the latest version of Pangolin by making it explicit on the PipeCoV sourcecode. All steps above can be run with the following command line, for example:

assembly_docker.sh -i illumina -1 hq_reads.fastq -2 hq_reads.fastq

-r sarscov2_MN908947.fasta -k 31 -m 2 -l 100 -c 10 -o output_assembly

-t 24 -s illumina_rtpcr

Help commands:

-i is the sequencing instrument;

-1 is the forward high-quality reads;

-2 is the reverse high-quality reads;

-r is the file that contains the reference genome;

-k is the a kmer-size;

-m is the maximum of mismatch;

-l is the selecting contigs bigger than a minimum length;

-c is the with higher coverage;

-o is the output folder;

-t is the number of threads to be used;

-s is the sample name;

To compare all the results from the benchmarked pipelines, we used boxplot graphs and all the statistics can be found in the Supplemental Material.

Results and discussion

From 210 sequencing runs present in Bioproject PRJNA691556 (Gupta et al., 2021), 120 were paired-end sequencing performed with the MiSeq Illumina platform, counting more than 72 Mb each. Hence, we benchmarked PipeCoV with those 120 paired-end datasets and compared the results to four other pipelines. All data relating to the 120 sequenced samples can be found in the Supplemental Material.

Figure 2A shows the average size of the consensus generated by each pipeline. We can observe that Signal and V-pipe generated an average size that is equal to the reference genome length (MN908947, 29,903 bp), which is consistent with the implemented reference assembly strategy. CLC and Viralrecon also use reference-based assembly strategies, but they obtained an average genome size that is different from the reference genome (29,840 bp). PipeCoV obtained smaller and more variable values for the consensus length (average of 29,754 bp) because it combines de novo and reference-based strategies for the assembly. The pipelines which use reference-based strategy assembly tend to force the resulting consensus to have the same length of the reference genome used. This approach ends up by filling regions of the genome with generic bases (N’s) that may not be present in the actual genome. In de novo strategies, there is no need to force the consensus to have the same reference length, resulting in a consensus that may be smaller or longer than the reference.

Figure 2 (A) Consensus sequence length generated by the pipelines. (B) The average genome coverage of the consensus sequence generated by the pipelines.

Regarding genome coverage, PipeCov had the best performance in comparison to the other pipelines, showing an average genome coverage of 97.01%, followed by V-pipe (94.94%). Figure 2B shows the average coverage of the consensus genome generated by the five pipelines. Despite the more variable consensus length presented by PipeCoV, resulting from the dual assembly strategy (Fig. 2A), coverage was consistently higher in PipeCoV (Fig. 2B). The different lengths can also be explained by the lower number of N’s in high-quality assemblies presented by Pipecov (Figs. 3A and 4).

Figure 3 (A) The number of consensus sequences generated by the pipelines that are considered high-quality consensus, according to Briones et al. (2020). (B) The X-axis represents all the 120 samples ordered by descending number of N’s obtained by PipeCoV. The Y-Axis represents the number of N’s. Colors represent different pipelines.

Figure 4 The number of N’s in the consensus sequences generated by each pipeline.

Another meaningful quality metric of the consensus sequences assembled by the pipelines is the number of N’s and the size of each generated consensus sequence. According to Briones et al. (2020), the number of N’s in high-quality assemblies must be less than 1% of the total consensus length and the length of the consensus must be greater than 29,000 bp. Values that do not follow those thresholds may compromise many of the phylogenies and evolutionary studies of the virus, resulting in erroneous variant classification (Briones et al., 2020). Figure 3A shows that of the 120 consensus sequences generated by PipeCoV, 57 had the minimum quality standard, being 1.54-fold better than the second-best pipeline, Signal, that generated 37 high-quality genomes. PipeCoV generated three consensus sequences with the length between 29.000 bp and 28.837 bp and 64 consensus sequences with more than 300 N’s. Signal, ViralRecon, V-Pipe, and CLC did not generate any consensus sequence with length less than 29.000 bp, but generated 83, 90, 111, and 118 consensus sequences with more than 300 N’s, respectively. Once again, the results indicate that when combining de novo and reference-based assembly strategies produced significantly more high-quality consensus sequences (see Supplemental Material in the “Benchmarking’’ tab).

Figure 3B shows that in general, PipeCoV generated consensus sequences with a smaller number of N’s compared to the other pipelines. It is possible to notice in Fig. 3B that for some samples, PipeCoV generated consensus with less than 100 N’s, indicating a better performance for high-quality assemblies.

Regarding the number of N’s in the consensus generated by the pipelines, Fig. 4 shows that PipeCoV also had a smaller average number of N’s compared to the other pipelines, delivering a reliable genome consensus. This fact is again associated with the combination of de novo and reference-based assembly strategies, and the use of GapCloser, that eliminated gaps filled with N’s in the assemblies. It is possible to notice that the variance of the amount of N’s on PipeCoV results is smaller than for the other pipelines. On average, PipeCov generated consensus sequences with 324 N’s, being 1.94-fold better than the second-best pipeline regarding the number of N’s generated (V-pipe), with an average of 631 N’s.

Figure 5 shows that the pipelines presented very similar results for the variant identification when analyzed with Pangolin. Despite the similarities, V-pipe was not able to identify one of the B.1.1.306 and B.1.195 variants. Likewise, CLC could not identify one of the B.1.1.8 and B.1 variants. When we look further to find some reasons that might justify why V-pipe and CLC could not identify those variants for some samples, we observed that the samples SRR13418728, SRR13418882, SRR13418697 and SRR13418761 (that respectively should be classified as the B.1.1.306, B.1.195, B.1.1.8 and B.1 variants) present a total of 29,903, 29,903, 14,965 and 18,938 number of N’s, respectively. This suggests that depending on the number of N’s that a consensus has, this might impact the variant identification. New variants are being routinely discovered, making it necessary for a systematic update of Pangolin databases. Pangolin is not used in the pipelines we benchmarked. We also implemented automatic updates of the Pangolin database in PipeCoV, eliminating the need to download new versions at each execution.

Figure 5 Variants identified by the pipelines.

The figure shows that all pipelines identified almost the same variants.

Although we did not have interest in compare the pipelines regarding running time, we registered the time obtained by PipeCoV, CLC, V-pipe, ViralRecon, and SIGNAL to analyze all the 120 samples as being 32 h 06 m, 31 h 10 m, 30 h 42 m, 19 h 52 m, and 1 h 53 m, respectively. SIGNAL was by far the fastest pipeline to analyze the whole benchmarking dataset, but as we have showed in the previous results, it presented a higher number of N’s (Fig. 4) and a lower coverage percent (Fig. 2B) than PipeCoV.

Conclusions

Considering the vast amount of genomic data being routinely produced for the Sars-Cov2 virus and the emergence of new strains, we decided to create a free and user-friendly pipeline capable of generating results quickly and reliably, with only two command lines. PipeCoV was robust and performed well when compared to the other state-of-the-art pipelines. The results show that PipeCoV assembled high-quality genomes while including a complete analysis of the virus variants. One main difference of PipeCoV is the dual assembly approach. The de novo assembly strategy also adopted by PipeCoV preserves the biological structure of the genome variation, such as indels and single/multiple nucleotide variants, avoiding missing information, which might happen in analysis using only reference-based assembly. Also, PipeCoV always uses the latest version of the lineages database from Pangolin, thus discarding the possibility of erroneous or outdated identification. High-quality consensus sequences are essential to allow robust and reliable studies for phylogeny and evolution, and PipeCoV showed to generate more high-quality consensus sequences than the other pipelines.

Additional information is available in the Supplemental Material.

Supplemental Information

Supplemental Information 1 Benchmarking results for all pipelines tested.

Click here for additional data file.

Additional Information and Declarations

Competing Interests

Author Contributions

Data Availability

Guilherme Oliveira is an Academic Editor for PeerJ. The authors declare that they have no competing interests.

Renato R. M. Oliveira conceived and designed the experiments, performed the experiments, analyzed the data, prepared figures and/or tables, authored or reviewed drafts of the paper, and approved the final draft.

Tatianne Costa Negri conceived and designed the experiments, performed the experiments, analyzed the data, prepared figures and/or tables, authored or reviewed drafts of the paper, and approved the final draft.

Gisele Nunes conceived and designed the experiments, authored or reviewed drafts of the paper, and approved the final draft.

Inácio Medeiros performed the experiments, analyzed the data, authored or reviewed drafts of the paper, and approved the final draft.

Guilherme Araújo conceived and designed the experiments, authored or reviewed drafts of the paper, and approved the final draft.

Fabricio de Oliveira Silva conceived and designed the experiments, authored or reviewed drafts of the paper, and approved the final draft.

Jorge Estefano Santana de Souza conceived and designed the experiments, authored or reviewed drafts of the paper, and approved the final draft.

Ronnie Alves conceived and designed the experiments, authored or reviewed drafts of the paper, and approved the final draft.

Guilherme Oliveira conceived and designed the experiments, authored or reviewed drafts of the paper, and approved the final draft.

The following information was supplied regarding data availability:

The source code is available at GitHub: https://github.com/alvesrco/pipecov.

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
