# Peer review of "PipeCoV: a pipeline for SARS-CoV-2 genome assembly, annotation and variant identification"

_PeerJ, doi:10.7717/peerj.13300_

## Round 0.1 · original submission · Major Revisions

· Academic Editor

Major Revisions

Dear corresponding author, this article was reviewed by three independent referees, and although they have appreciated the manuscript, there are essential corrections that deserve further attention, including minor revisions. But in particular, those pointed out by reviewer 3, which I consider to be of more significant concern.

·

Basic reporting

I reviewed with interest the manuscript entitled " PipeCoV: a pipeline for SARS-CoV-2 genome assembly, annotation and variant identification". PipeCov is a good example of intelligent and well-organized pipeline to assembly Sars-CoV-2 genomes, combining the advantages of using de novo and reference−based assembly strategies. As the authors demonstrate in detail throughout the manuscript, the use of these combined strategies allows to improve the assembly parameters with respect to the current reference assemblers for SARS-CoV-2. Another important aspect of PipeCov is it´s unique feature to have the pangolim automatically updated, eliminating the need to download new versions at each execution.

Experimental design

The methods are well documented and suitable for the present work

Validity of the findings

Minor comments:
Line 133 page 5. It seems like there are a couple of wrong characters (!,?)

Reviewer 2 ·

Basic reporting

The PipeCov, a pipeline for SARS CoV-2 genome assembly, annotation, and variant identification, has an excellent approach for analyzing the novel coronavirus (SARS-CoV-2) with promising results. As they compare with other pipelines, they choose to use a "hybrid" approach with de novo and reference genome assemblies. They took advantage of these two procedures to minimize Ns' presence in the consensus sequences while maximizing the average coverage.
However, some issues need to be clarified to understand how the pipeline works.

Experimental design

About the methods, I missed the information about the default values of the pipeline. Honestly, are there any default values? I did not see it explicitly in the text. For example, what is the meaning of "low Phread score" (l.86) ? Which is the threshold of low quality and sequence length?
Yet, I do not understand the explanation in figure 1 how the "de novo" assembly with Spades connects with the reference genome of the Bowtie and Samtools. It seems this connection emerges by applying the script hcov_generate_consensus.R. I think it is worth a brief explanation of how it works despite the reference.

Validity of the findings

About the results, I am wondering why "120 amplicon datasets randomly extracted from 210 samples under "just one" Bioproject? Why not test in a few other Bioprojects? Another issue is how PipeCov keeps Pangolim automatically updated without downloading a new version at each execution?

Additional comments

In figure 5, the X axis is missing. It was not explained in the legend.

Reviewer 3 ·

Basic reporting

First of all I suggest to review the english, spite of not being an english native person, I found some phrases a bit strange to interpret.
Then, I do not find the reason to promote the development of the pipeline. What is the problem that they intend to solve? Whay this is important? Why the reference and de-novo are required in this context? The emergence of the need is not clearly exposed.
The Table with sample details should be given as supplementary material and should be in english. It is not referenced on the text.

Experimental design

A section explaining the comparison process and used statistics as well as the rational of comparison procedures should be included in the methods section. The rational of the statistical methods should be provided.
For instance, results of figure 2 and 3 deserve some statistical test as well as to represent results as a boxplots to evaluate real differences between approaches.
Since the Briones et al 2020 is a technical and not peer reviewed manuscript. Spite that it can be ok, what they claim, I suggest that in this manuscript the quality rational should be stated as well as why the 1% Ns limits and the quality consensus and the explanation of the quality standards with appropriate discussion.

Figure 5, I do not understand the x-axis.

Validity of the findings

the novelty and needs for the development were not clearly exposed. Thus conclusions are weak.

---

## Round 0.2 · Minor Revisions

· Academic Editor

Minor Revisions

Basic report:
Spite the referee considering that having good coverage is a desirable characteristic and having low Ns inserted in the genome, it seems that any of these advantages impact variant discovery. Still, the referee said the authors claimed that PipeCov would significantly affect phylogeny analysis, but this is not demonstrated nor evaluated, thus being just a hypothesis.

Additional comments
Minor revisions:
The referee thinks that a speed comparison should be included to evaluate the advantages of paying time efforts in the analysis if the intention is, for instance, to look for variants.

line 192:
"of N's compared to the other pipelines. It is possible to no notice in Figure 3B that for some samples," the referee thinks it should be of N's compared to the other pipelines. It is possible to notice in Figure 3B that for some samples.

Reviewer 2 ·

Basic reporting

The PipeCov, a pipeline for SARS CoV-2 genome assembly, annotation, and variant identification, has an excellent approach for analyzing the novel coronavirus (SARS-CoV-2) with promising results. As they compare with other pipelines, they choose to use a "hybrid" approach with de novo and reference genome assemblies. They took advantage of these two procedures to minimize Ns' presence in the consensus sequences while maximizing the average coverage.
After modifications, it is clear to understand how the pipeline works

Experimental design

Methods is described with enough details to understand the scope and the aims of the pipeline

Validity of the findings

The object of the pipeline is well stated. The results is promising for the analysis of the virus SARS Cov-2 included those for de novo and reference genome.

Additional comments

No additional comments

Reviewer 3 ·

Basic reporting

The manuscript has been imrpoved, spite there are still some things that are not clear enouph for me.


Spite I consider that having good coverage is a desirable characteristic as well as having low Ns inserted in the genome, it seems that any of these advantage have an impact in variant discovery. The authors claimed that PipeCov will have a mayor impact on phylogeny analisis, but this is not demosntrated nor evaluated thus being just an hypothesis.

Experimental design

no comment

Validity of the findings

no comment

Additional comments

Minor revisions:
So, I thhink that a speed comparison should be included in order to evaluate the advantages of paying time efforts in the analysis if the intention is, for instance, to look for variants.

line 192:
"of N’s compared to the other pipelines. It is possible to no notice in Figure 3B that for some samples,"
i think that it should be

of N’s compared to the other pipelines. It is possible to notice in Figure 3B that for some samples,

---

## Round 0.3 · accepted · Accept

· Academic Editor

Accept

Dear Dr. Oliveira,

Considering that the questions raised by the referees were answered with precision and criteria, the article is robust to be published in PeerJ.
Best Regards

---

## Author Rebuttal · Round 0.3

# Response to Reviewer's comments

Dear Marisa Nicolas and Reviewers, we would like to thank you again for all the comments and suggestions that you made regarding our research and we are glad to satisfy all the requests made by Reviewer 2 and almost all of the requests made by Reviewer 3.

Here follow the last comments made by Reviewer 3 and how we addressed them in the new version of the manuscript.

## Reviewer 3:

**- Spite I consider that having good coverage is a desirable characteristic as well as having low Ns inserted in the genome, it seems that any of these advantage have an impact in variant discovery. The authors claimed that PipeCov will have a mayor impact on phylogeny analisis, but this is not demosntrated nor evaluated thus being just an hypothesis.**

- We included an observation in the manuscript. (Lines 203-208)

**- So, I thhink that a speed comparison should be included in order to evaluate the advantages of paying time efforts in the analysis if the intention is, for instance, to look for variants.**

- Although we did not have interest in compare the pipelines regarding time, we included this information in the manuscript. (Lines 212-216)

**- line 192:**

**"of N's compared to the other pipelines. It is possible to no notice in Figure 3B that for some samples,"**

**i think that it should be**

**of N's compared to the other pipelines. It is possible to notice in Figure 3B that for some samples,**

-We corrected in the manuscript. (Line 192)